# “Balance Is Better”: The Wellbeing Benefits of Participating in a Breadth of Sports across a Variety of Settings during Adolescence

**DOI:** 10.3390/ijerph19148597

**Published:** 2022-07-14

**Authors:** Oliver W. A. Wilson, Chris Whatman, Simon Walters, Sierra Keung, Dion Enari, Alex Chiet, Sarah-Kate Millar, Lesley Ferkins, Erica Hinckson, Jeremy Hapeta, Michael Sam, Justin Richards

**Affiliations:** 1Te Hau Kori, Faculty of Health, Victoria University of Wellington, Wellington 6012, New Zealand; oliver.wilson@vuw.ac.nz; 2School of Sport & Recreation, Auckland University of Technology, Auckland 1142, New Zealand; chris.whatman@aut.ac.nz (C.W.); simon.walters@aut.ac.nz (S.W.); sierra.keung@aut.ac.nz (S.K.); dion.enari@aut.ac.nz (D.E.); lesley.ferkins@aut.ac.nz (L.F.); erica.hinckson@aut.ac.nz (E.H.); 3Sport New Zealand, Wellington 6140, New Zealand; alex.chiet@sportnz.org.nz; 4Faculty of Health, University of Canterbury, Christchurch 8041, New Zealand; sarah-kate.millar@canterbury.ac.nz; 5School of Physical Education, Sport and Exercise Sciences, University of Otago, Dunedin 9054, New Zealand; jeremy.hapeta@otago.ac.nz (J.H.); mike.sam@otago.ac.nz (M.S.)

**Keywords:** physical activity, sport, exercise, recreation, leisure, well-being, happiness, adolescents, young people, coach

## Abstract

The purpose of this study was to examine how wellbeing is associated with the setting in which sport participation takes place and the breadth of sport participation. Demographic characteristics (age, gender, ethnicity, deprivation, (dis)ability status), recreational physical activity, and wellbeing were assessed in cohorts of adolescents (11–17 years) between 2017 and 2019 in Aotearoa, New Zealand. Better wellbeing was associated with participation in any sport vs. none (OR = 1.57, 95% CI = 1.30–1.90). Better wellbeing was also associated with participating in any coached sport training (OR = 1.48, 95% CI = 1.33–1.66), competitive sport (OR = 1.33, 95% CI = 1.18–1.49), social sport (OR = 1.33, 95% CI = 1.18–1.49), and uncoached sport training (OR = 1.16, 95% CI = 1.03–1.31) compared to non-participation in the given setting. Wellbeing was not associated with participation in physical education or solo sport. Participating in sport in three to five different settings (3 settings: OR = 1.21, 95% CI = 1.01–1.44; 4 settings: OR = 1.33, 95% CI = 1.09–1.62; 5 settings: OR = 1.37, 95% CI = 1.07–1.75) or sports (3 sports: OR = 1.25, 95% CI = 1.04–1.51; 4 sports: OR = 1.31, 95% CI = 1.06–1.61; 5 sports: OR = 1.33, 95% CI = 1.05–1.69) was associated with better wellbeing compared to participation in a single setting or sport, respectively. A balanced approach to participating across a variety of sport settings and sports that are facilitated by quality coaches may offer the largest additional wellbeing value.

## 1. Introduction

Improving population-level wellbeing is acknowledged as a priority worldwide [1,2,3]. Aotearoa, New Zealand (NZ), is no exception, with NZ’s Living Standards Framework (LSF) informed by several decades of international wellbeing research [4]. The LSF details 12 domains of current wellbeing, along with the roles that institutions and organisations have to play in wellbeing promotion [5]. Participation in physical activities and sports during adolescence is associated with numerous wellbeing domains outlined in the LSF and the broader global development agenda [6]. There is also a growing body of evidence that indicates quality sport experiences can offer wellbeing benefits above and beyond those associated with participation in physical activity in general [7,8]. However, large epidemiological studies that explore the mechanisms that underpin these findings are currently lacking.

There is emerging evidence suggesting that the association between sport participation and some aspects of wellbeing may be moderated by certain characteristics of different sports. Participating in different sports and settings provide people with varying opportunities to satisfy their need for autonomy, competence, and relatedness [9]. For example, the level of interdependence and the relative focus on aesthetic appearance in different sporting environments both appear to have implications on the social and psychological impact of participation [7,10]. Indeed, multiple mechanistic pathways that link physical activity with wellbeing outcomes have been described and the context in which the participation takes place was identified as a key moderator [11].

Understanding the intricacies of the association between the breadth of sport participation, both in terms of the number of sports and sport settings, and wellbeing is important. Evidence suggests that when youth narrow in on participating in a particular sport at the expense of participation in other sports, there is increased risk of injury [12,13,14,15,16] and burnout [14,15,17,18], and that this narrow focus can interfere with social relationships [14,15]. Beyond the direct implications of injury, burnout, and compromised social relationships, these all also happen to be antecedents to dropout from sport [14,19]. By contrast, evidence suggests that participation across a variety of sports tends to decrease the risk of injury and burnout relative to young people who have a narrow sporting focus [12,16,18]. Avoidance of a narrow sporting focus (specialization) and participation in multiple sports (sampling) are components of the “Balance is Better” philosophy espoused by Sport New Zealand Ihi Aotearoa (Sport NZ), NZ’s crown entity responsible for physical activity promotion [20].

To date, there has been limited research that has examined the unique association between wellbeing and sports participation across multiple settings (i.e., contexts) simultaneously, or the actual dose-response association between the number of sports participated in and the overall wellbeing of young people. In this paper, we focus on exploring the wellbeing value of different characteristics of sport participation. Our aim is to examine how known aspects of wellbeing are associated with: (i) the settings in which sport participation takes place; (ii) the breadth of sport participation. Specifically, we interrogate the relative wellbeing benefit of participating in sport in different contexts as well as the value of participating across multiple settings and multiple sports. In doing so, we generate policy- and practice-relevant insights for promoting participation in sport in ways that will optimize its impact on the wellbeing of young people.

## 2. Materials and Methods

### 2.1. Participants and Procedures

Data were collected as a part of the Active NZ Young Peoples survey Sport New Zealand Ihi Aotearoa [21]. Data included in the current study were collected continuously from the beginning of 2017 to the end of 2019. Young people, children, and adolescents aged 5–17 years at baseline, were recruited via adults residing in their household who were identified to participate in the Active NZ adults survey using the NZ electoral roll as a sampling frame. Full survey methods, including details of the sample frame, are described in the annual Active NZ Technical reports [22,23,24]. Respondents who did not have complete socio-demographic characteristics, physical activity, and wellbeing data were excluded. Those younger than 11 years and those no longer at school were also excluded. Analyses were conducted on 6771 young people.

### 2.2. Measures

#### 2.2.1. Demographic Characteristics

Age: Respondents identified their age in years.

Gender: Respondents identified their gender (male, female, or gender diverse). Due to limited sample size for gender diverse our inferential analyses focused on cis-gender individuals.

Ethnicity: Respondents identified their ethnic group(s) and there was no limit on the number of ethnicities they could choose. For the purposes of these analyses, respondents who identified multiple ethnicities were categorised to only one ethnic group using the following prioritisation: Māori, Pasifika, Asian, Middle Eastern/Latin American/African (MELAA), European, Other. These ethnic groups were selected based on those specified by Statistics NZ. Due to limited sample size for Other ethnicities, our inferential analyses did not include this group.

Disability status: Respondents who did not report using a wheelchair, using a walking aid, using prosthetics, or dealing with an ongoing physical illness were classified as someone without a disability.

Deprivation status: Deprivation was determined using the 2018 NZ Index of Deprivation, which combines census data relating to income, home ownership, employment, qualifications, family structure, housing, access to transport, and communications to designate small geographic areas (60–110 people) with a decile number ranging from 1 (least deprived) to 10 (most deprived) [25]. Respondents were classified as residing in low (deciles 1–3), medium (deciles 4–7), and high (deciles 8–10) deprivation areas.

#### 2.2.2. Physical Activity and Sport Participation

Participation: Respondents were asked whether they had participated in any physical activity that was specifically for the purpose of sport, exercise, or recreation in the past seven days (yes/no). Those who answered yes were classified as participants in “recreational physical activity” and were then asked to identify from a list of 77 options which activities they participated in during the past seven days. There was also an “other” option provided with free text for respondents to describe any activity they had done that was not listed.

Physical activity and sport classification: The list of recreational physical activities included “non-sport recreational activity” (e.g., tramping or bush walks, gym), as well as a range of sports. For the purpose of this study, the following activities were considered “sport”: Adventure racing, athletics, badminton, basketball, body boarding, boxing, canoeing or kayaking, cheerleading, cricket, croquet, cross country, cycling or biking, dance/dancing, football/soccer, futsal, golf, gymnastics, handball, hockey or floorball, indoor climbing, jiu jitsu, ki-o-rahi, kapa haka, karate, mountain biking, motorbiking, motocross, netball, orienteering, paddle boarding, parkour, rock climbing, rollerblading, roller skating, rowing, rugby or rippa rugby, rugby league, running/jogging, sailing or yachting, scuba diving, scootering, skateboarding, skiing, snowboarding, softball, squash, surf lifesaving, surfing, swimming, table tennis, tae kwon do, tennis, touch, trampoline, triathlon or duathlon, ultimate frisbee, volleyball, waka ama, wake boarding, water polo or flippa ball, water skiing. Further information about participation in different sports can be accessed online [26].

Setting: For activities that they had participated in, respondents were asked in what settings they had participated: ‘In PE or class at school’ (physical education); ‘In a competition or tournament’ (competitive sport); ‘Training or practicing with a coach/instructor’ (coached sport training); ‘Playing or hanging out with family or friends’ (social sport); ‘Playing on my own’ (solo sport); ‘For extra exercise, training, or practice without a coach or instructor’ (uncoached sport training).

Duration: If respondents indicated that they had participated in a given activity in a given setting, they were asked how long they participated in that activity/setting each week (15 min, 30 min, 45 min, 1 h, 1.5 h, 2 h, 3 h, 4 h, 5 h+). These were summed across all settings and/or activity types to calculate total durations as a continuous variable for the analyses. Respondents who identified any activity and provided information about setting and duration were classified as a participant in that activity type for categorical analyses (i.e., vs. non-participant).

#### 2.2.3. Wellbeing

Respondents were asked to respond to a question rating their wellbeing on a 10-point scale ranging from 1 (very unhappy) to 10 (very happy). Whilst it is recognized that wellbeing is a multi-dimensional construct, the single item measure was used in this study because it has been shown to be a valid overall wellbeing indicator and aligns with the OECD Guidelines on Measuring Subjective Well-being [27]. Based on the distribution of the data, those whose response was ≥8 were categorized as having “better wellbeing”.

### 2.3. Statistical Analyses

Analyses were conducted using SPSS (Version 28.0). Descriptive statistics were computed to describe the sociodemographic characteristics of the sample and the physical activity participation characteristics according to setting and number of sports. The associations between setting/sport participation and wellbeing were examined using a series of binary logistic regression analyses. Firstly, the association between wellbeing and sport participation in each of the six settings was examined in two ways: (i) participation in any vs. no sport (i.e., categorical models); (ii) participation in each additional hour/week of sport (i.e., continuous models). Secondly, the association between wellbeing and participating in any vs. no sport across all settings was examined. Thirdly, for those participating in sport, the association between wellbeing and their breadth of participation was examined in two ways: (i) participation in each additional setting; (ii) participation in each additional type of sport. All analyses were initially conducted using raw data to calculate crude odds ratios (ORs), and subsequent models were adjusted for socio-demographics and non-sport recreational activity (i.e., details of the covariates included in each model are specified in the results tables). We calculated 95% confidence intervals (CIs) for all of the ORs reported and used these to assess statistical significance (i.e., 95% CIs not crossing 1.0 is equivalent to *p* < 0.05).

## 3. Results

### 3.1. Participant Characteristics

The sociodemographic characteristics of the sample have been described in detail elsewhere [8]. In summary, the sample was evenly split between males and females, the majority were European (70.5%), were without a physical disability (94.6%), and resided in low-mid-deprivation areas (81.8%). Most respondents reported participation in recreational physical activity in the past seven days (94.7%), spending an average of 10.9 ± 10.1 hrs/wk participating. Of this, an average of 7.6 ± 7.6 hrs/wk was spent doing sport.

### 3.2. Sport Setting and Wellbeing

Descriptive statistics for sport participation in different settings and the associations with wellbeing are reported in Table 1. Participation levels were highest in terms of proportion participating and mean duration for social sport (57.3%; 1.9 hrs/wk) and coached sport training (55.6%; 1.9 hrs/wk). Uncoached sport training had the lowest participation levels (32.8%; 0.7 hrs/wk). Based on the fully adjusted models, the association between sport participation and wellbeing varied according to setting. Better wellbeing was associated with participating in any coached sport training (48% higher odds), competitive sport (33% higher odds), social sport (33% higher odds), and uncoached sport training (16% higher odds). Each additional hour of coached sport training (13% higher odds), competitive sport (8% higher odds), social sport (5% higher odds) was also associated with better wellbeing. There was no association between wellbeing and participation in physical education or solo sport.

### 3.3. Breadth of Sport Participation and Wellbeing

Descriptive statistics regarding the breadth of participation based on number of settings and sports, along with the associations between participation breadth and wellbeing are reported in Table 2. A total of 91.2% of the sample participated in sport and of those, over half participated in 2–4 different settings (61.0%) and 2–4 different sports (50.2%). Compared to non-participants, those doing any sport had 57% higher odds of reporting better wellbeing. Despite no statistically significant difference between participating in one or two settings, compared to participating in only one setting, the odds of having better wellbeing were greater for those participating in three (21% higher odds), four (33% higher odds), or five (37% higher odds) settings. The odds of having better wellbeing from participating in all six settings were not statistically significant. Similarly, while there was no statistically significant difference between participating in one or two different sports, compared to participating in only one sport, the odds of having better wellbeing were greater for those participating in three (25% higher odds), four (31% higher odds), or five (33% higher odds) sports. The odds of having better wellbeing from participating in six or more sports were not statistically significant.

## 4. Discussion

Our results indicate that most adolescents in NZ participate in sport on a weekly basis and do so across multiple settings and sports. Building upon previous work demonstrating the unique additional wellbeing value of sport participation [8], findings from the current study suggest that this relationship varies according to the setting, duration, and breadth of sport participation during adolescence.

Self-determination theory offers a useful framework to interpret differences in the association between sport participation and wellbeing across different settings. Self-determination theory posits that behaviour is driven by the desire to satisfy three fundamental human needs (autonomy, competence, and relatedness [9]), which are key to wellbeing [28,29]. When it came to specific settings, the strongest association was observed in relation to coached sport training, followed by competitive sport, and social sport. Each of these settings offers an opportunity for young people to satisfy one or more of the aforementioned needs [30,31].

Participation in neither physical education nor solo sport were found to be associated with wellbeing in our study. These findings may also be explained by self-determination theory, in that these activities may restrict young people from satisfying needs for autonomy, competence, and/or relatedness [29]. For example, adolescents have no choice (i.e., autonomy) as to whether they participate in physical education until reaching year 11 (the third to last year of secondary school) in NZ, and the activities that they participate in as a part of physical education may not be desirable or offer them a chance to demonstrate competence to themselves and/or others. Though adolescents would likely be exercising autonomy to participate in solo sport, opportunities to satisfy the needs for relatedness or competence would typically be lower relative to other settings. This lack of relatedness and external validation of competence may also explain our results for uncoached sport training, which had an association with wellbeing that was weaker than activities involving a coach and that was independent of participation duration.

In contrast, the duration of coached sport training, competitive sport, and social sport was positively associated with wellbeing. This positive dose-response relationship is consistent with the current physical activity recommendations, which do not set an upper limit on participation duration and indicate that there is additional benefit from participating in more physical activity provided it is incrementally increased [32]. However, there is evidence from within the sport sector suggesting that there is an upper limit to this relationship, particularly in regard to the risk of injury, and that this may vary according to sport and setting [33].

Our results also indicate that the breadth of sport participation is important for wellbeing. When compared to non-participants, the respondents who participated in any sport had better wellbeing outcomes. Importantly, this association strengthens as the number of sports and settings increases up to a certain point when the wellbeing benefits appear to plateau and potentially drop. This suggests that there is a “sweet spot” of participating in three to five sports and/or settings where optimal wellbeing outcomes are realized at a population level.

### 4.1. Implications

The variation in wellbeing value of sport participation in different settings clearly demonstrates the importance of developing quality coaches that provide positive experiences through the delivery of youth sport. Our findings are consistent with evidence indicating that coaches and the climate they create play an important role in promoting adolescent wellbeing [34,35]. In contrast, the lack of association observed between physical education and wellbeing may reflect the variable quality of physical education delivery within schools at the time of data collection. In recent years, NZ physical education has been characterized as inconsistent and variable [36]. Our results may reflect the changes in initial teacher education training, pedagogical approaches, and curriculum adherence that have occurred as physical education specialists within schools have been gradually replaced by external providers in the past 10–15 years [37,38]. However, these issues are now widely recognized and have led to the development of the Healthy Active Learning initiative, which is a cross-government partnership aimed at improving the wellbeing of young people in NZ through quality physical activity in schools [39].

The additional wellbeing value of participating in several different sports and across a variety of settings strongly aligns with the notion that “Balance is Better” [20]. Our findings are consistent with existing evidence against specialisation in one sport during adolescence [12,16,18]. Therefore, our findings support the need to ensure that the governance, funding, and structures within the sport sector are aligned in a way that enables young people access to a variety of physical activity opportunities throughout adolescence and beyond. Our results also demonstrate that the breadth of sport participation can have immediate wellbeing benefits that may be of interest to policy-makers who have historically focused on youth participation because of its association with future physical activity levels during adulthood [40]. We hypothesize that the immediate benefits of a more balanced approach to participation in sport during adolescence extends to wellbeing outcomes beyond the reduction of injury risk and that this warrants further investigation.

Finally, this study has highlighted the importance of context in the relationship between physical activity participation and the wellbeing of young people. In this regard, our findings were restricted to the physical activity types and settings that were articulated in the available data. We recommend broader investigation into how different intra-personal, inter-personal, socio-cultural, and environmental determinants of physical activity behaviour may influence the subsequent wellbeing outcomes [41].

### 4.2. Limitations

Firstly, our data included very few “overly active” young people (i.e., those at the upper end of the dose-response curve that engage in a very large volume of sport and/or number of sports). Existing evidence indicates that there is an upper limit beyond which wellbeing will be negatively impacted by more participation, but we were unable to interrogate this within the limitations of our data [33]. Future research on “overly active” young people may require more targeted sampling and the use of assessment methods that are more bespoke than those applied in this population level survey.

Secondly, the cross-sectional nature of this study means it is not possible to determine the direction of causation for the associations we have described between sport participation and wellbeing. Although there is substantial evidence concerning the impact of physical activity participation on wellbeing in both children and youth [7,11] and adults [42], a reciprocal relationship is probable. Specifically, wellbeing may be an antecedent to sport participation as well as a consequence of participation. Thus, in addition to advocating for sport to promote youth wellbeing, improving youth wellbeing using other mechanisms may also directly contribute to increasing their participation in sport. It is also important to acknowledge that this reciprocal relationship may vary according to sport, setting, and breadth of participation. Further longitudinal research is warranted to better understand the temporal association between sport participation and wellbeing, whether these associations are enduring or fleeting, and the mechanisms underpinning this presumably reciprocal relationship.

Finally, there are limitations pertaining to the measurement of certain variables. With respect to physical activity, the level (i.e., level of competitiveness) of sport participation was unknown, and measuring participation duration does not necessarily account for the intensity physical activity. However, given our focus was on the activity setting and the number of sports, self-report methods are the most pragmatic and valid way to collect data from an adequate sample. Items used to measure disability focused primarily on mobility and did not encompass sensory or intellectual disabilities, which require consideration in future research. Finally, there is ongoing conjecture about the definition and measurement of wellbeing in the international literature, and this continues to evolve. Despite this, there is good evidence to suggest that the single survey item used in this study provides a robust, albeit blunt, indicator for overall wellbeing [27]. However, we acknowledge that the measure used in our study provides limited insight into the various domains of wellbeing, and may not adequately capture indigenous perspectives. Thus, a more nuanced approach to investigating the contribution of sport participation to each of these domains is warranted across different population groups.

## 5. Conclusions

In summary, participation in a variety of sports and sport settings, particularly those involving coaches, appear to have the strongest association with wellbeing. Our results align with the “Balance is Better” philosophy, in that spreading participation across a broader number of settings and sports is positively associated with wellbeing within a certain “sweet spot”. There are some signs that this relationship may reverse or deteriorate beyond participating in five sports and/or settings, but further research is required to substantiate these findings using targeted sampling procedures, more nuanced measures, and stronger study designs.

## Figures and Tables

**Table 1 ijerph-19-08597-t001:** Association between participation in the various sports settings and wellbeing (n = 6725).

	Categorical				Continuous			
	n	%	Model 1	Model 2	M	SD	Model 1	Model 2
			OR (95% CI)		(hrs/wk)		OR (95% CI)	
Physical education	3581	53.2	**1.17 (1.05–1.30)**	0.96 (0.86–1.08)	1.3	2.1	1.01 (0.98–1.04)	0.98 (0.95–1.01)
Competitive sport	2400	35.7	**1.34 (1.19–1.50)**	**1.33 (1.18–1.49)**	1.0	2.0	**1.08 (1.04–1.11)**	**1.08 (1.04–1.11)**
Coached sport training	3740	55.6	**1.59 (1.42–1.77)**	**1.48 (1.33–1.66)**	1.9	2.5	**1.13 (1.11–1.16)**	**1.13 (1.10–1.16)**
Social sport	3853	57.3	**1.59 (1.43–1.78)**	**1.33 (1.18–1.49)**	1.9	3.1	**1.08 (1.06–1.10)**	**1.05 (1.03–1.07)**
Solo sport	2943	43.8	1.06 (0.95–1.19)	0.99 (0.89–1.11)	0.9	1.8	1.02 (0.99–1.06)	1.01 (0.98–1.04)
Uncoached sport training	2203	32.8	0.96 (0.86–1.08)	**1.16 (1.03–1.31)**	0.7	1.6	0.94 (0.91–0.98)	0.99 (0.95–1.03)

Notes. Statistically significant odds ratios are bolded. In each model predictor variables are included simultaneously (i.e., in a single mode l); referent variables in the categorical models are no participation in the given setting; participation duration (hrs/wk ) was included as the predictor in the continuous model; Model 1—crude unadjusted; Model 2—fully adjusted—gender, age (years), ethnicity, deprivation, disability status, and non-sport recreational activity (any vs. none in categorical model; hrs/wk in continuous model) included as covariates.

**Table 2 ijerph-19-08597-t002:** Association between sport participation breadth and wellbeing (n = 6725).

		n	%	Model 1	Model 2	Model 3
				OR (95% CI)		
Any sport	No sport (referent)			1.00	1.00	1.00
	Any sport	6133	91.2	**2.66 (2.24–3.16)**	**2.08 (1.74–2.48)**	**1.57 (1.30–1.90)**
Breadth of sport setting	1 setting (referent)	952	14.2	1.00	1.00	1.00
	2 settings	1450	21.6	**1.37 (1.16–1.61)**	**1.25 (1.06–1.49)**	1.17 (0.99–1.39)
	3 settings	1501	22.3	**1.59 (1.35–1.88)**	**1.39 (1.17–1.64)**	**1.21 (1.01–1.44)**
	4 settings	1153	17.1	**1.98 (1.65–2.37)**	**1.64 (1.36–1.97)**	**1.33 (1.09–1.62)**
	5 settings	709	10.5	**2.34 (1.90–2.89)**	**1.89 (1.53–2.35)**	**1.37 (1.07–1.75)**
	6 settings	368	5.5	**2.69 (2.05–3.53)**	**2.25 (1.70–2.97)**	1.34 (0.96–1.86)
Breadth of sport type	1 sport (referent)	865	12.9	1.00	1.00	1.00
	2 sports	1226	18.2	**1.25 (1.05–1.49)**	1.17 (0.98–1.41)	1.10 (0.91–1.32)
	3 sports	1221	18.2	**1.64 (1.37–1.96)**	**1.41 (1.18–1.70)**	**1.25 (1.04–1.51)**
	4 sports	930	13.8	**1.90 (1.57–2.30)**	**1.58 (1.30–1.93)**	**1.31 (1.06–1.61)**
	5 sports	674	10.1	**2.25 (1.82–2.79)**	**1.71 (1.37–2.13)**	**1.33 (1.05–1.69)**
	6+ sports	1217	18.1	**2.34 (1.95–2.82)**	**1.82 (1.50–2.20)**	1.22 (0.97–1.53)

Notes. Statistically significant odds ratios are bolded. Model 1—crude unadjusted; Model 2—fully adjusted—gender, age (years), ethnicity, deprivation, and disability status included as covariates; Model 3—gender, age (years), ethnicity, deprivation, disability status, total recreational physical activity (hrs/wk ) included as covariates.

## Data Availability

Publicly available datasets were analyzed in this study. These data can be provided on request from research@sportnz.org.nz.

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
