# Peer review of "“Balance Is Better”: The Wellbeing Benefits of Participating in a Breadth of Sports across a Variety of Settings during Adolescence"

_ijerph, 2022, doi:10.3390/ijerph19148597_

Round 1

Reviewer 1 Report

In this manuscript, a cross-sectional study is presented in which associations between sport participation in multiple settings and wellbeing were examined in a large sample of adolescents in New Zealand. The topic falls squarely within the domain covered by this journal. The large sample size and the high quality of the writing are desirable features of the manuscript. These positive impressions notwithstanding, I have several concerns about the manuscript in its current form:

1.         Why was the continuous variable of deprivation status (described on lines 108-113) converted into a categorical variable?

2.         Why was such a small window of time used for assessing sport participation? What if participants were injured, ill, or “out of season” during the assessment window?

3.         It should be noted that as assessed in the current study, duration doesn’t necessarily correspond to the intensity or actual amount of physical activity experienced by participants.

4.         The “type of sport” analysis described on l. 166-167 fails to recognize that multi-sport involvement is often sequential by season and is, therefore, not capturable in a 7-day assessment window. That said, it is nothing short of astonishing that 18% of the sample reported participating in 6 or more sports in a 7-day period (as shown in Table 2). Is it reasonable to expect children to have access to so many different sport options?

5.         Although the cross-sectional research design is duly noted as a limitation of the study in section 4.2, the bulk of the Discussion section is written exclusively from the perspective of sport participation serving as an antecedent of wellbeing. As acknowledged on l. 311, “wellbeing may be an antecedent of sport participation.” This angle deserves further consideration in the Discussion section and perhaps even the Introduction section.

6.         It would make sense to advocate more strongly for longitudinal research on the topic under consideration. Longitudinal studies are needed not only to establish time-order relationships among the primary variables of interest, but also to find out if the associations between sport participation and wellbeing are enduring or only fleeting.

Reviewer 2 Report

We congratulate the authors of the presented paper and, in a constructive spirit, we allow ourselves to suggest a list of aspects that we hope can contribute to improving its quality.

If they base the interpretation of the results on the self-determination theory, it would be convenient to include a reference to it in the introduction.

For a better understanding of the representativeness of the sample, it would be desirable to indicate the sample universe.

When they state the "Disability status", they only refer to motor disability, it would be convenient to justify why they do not include another type of disability, such as intellectual disability, or at least refer to it in the limitations, since it can affect the perception of well-being.

They refer that scores equial to or greater than 8 on the well-being scale were categorized as "better well-being", but they do not indicate whether other categories could be established for the rest of the values of the scale, which could enrich the results obtained.

In line 207, they mention that most of the participants do it in 2-4 different sports, but the data in table 2 indicate that they are only half. We recommend rephrasing that statement.

There is an error in the first bibliographic reference (line 353), since a parenthesis is opened indicating having accessed, but not the date, and the parenthesis is not closed either

Finally, it would be desirable to establish the influence of practicing sports at an elite level on the possibility of practicing other sports or in other settings. We recommend including it in future research.

Reviewer 3 Report

Please provide the results, if possible, a combination of which sports (specific names of disciplines described in lines 125-134 were most often chosen by the study participants, and discuss what could have resulted from it to the social needs of educational sports programs.
